# Study on the Evolutionary Game of Cooperation and Innovation in Science and Technology Town Enterprises

Feng Li and Yalong Wang *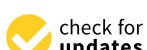

School of Economics and Management, Beijing Information Science and Technology University,
Beijing 100192, China; li-feng0620@126.com
* Correspondence: wangyalong@bistu.edu.cn

**Abstract:** Developing science- and innovation-based special towns plays a pivotal role in new urbanization, and enterprise cooperative innovation can help accelerate the development of high-quality science- and innovation-based special towns. A dynamic evolutionary model of enterprise cooperative innovation under two different mechanisms of market law and government regulation is developed for a government-led science and innovation town. The influence of various influencing factors, including willingness to cooperate, on the choice of enterprise cooperative innovation strategy is examined, and MATLAB simulation is used to verify the analysis and provide suggestions for promoting the sustainability of enterprise cooperative innovation. As a result of the study, it is shown that factors influencing the choice of cooperative innovation strategy for enterprises under the market mechanism include the cost and benefits of cooperative innovation, the degree of willingness to cooperate, and the degree of effort. When the market mechanism fails, government subsidies and incentive bonuses are more effective than either alone when it comes to encouraging firms to adopt cooperative innovation strategies. When government penalties are lower than free-rider benefits, they cannot influence firms' speculative behavior. It is found that only a penalty intensity above the threshold can effectively inhibit the phenomenon of free-riding and motivate enterprises in science- and innovation-oriented characteristic towns to choose a cooperative innovation strategy, and the greater the penalty intensity, the faster the enterprises will tend to cooperate and innovate.

**Keywords:** science and technology town; cooperative innovation; evolutionary game theory; stability analysis

## 1. Introduction

Innovation is the first driving force of development, providing the strategic support necessary for the creation of a modern economic system. As General Secretary Xi Jinping pointed out, "innovation-driven development is one of the main forces in our development, which is the group army, which sometimes relies on smaller and medium-sized enterprises, "a trick", in order to support the innovation development of these enterprises". The construction and development of science- and innovation-oriented characteristic towns is precisely a concrete measure to implement this development strategy. It reflects the interaction between China's new urbanization and emerging industries [1], emphasizing the development of high precision by relying on specific industries, and focusing on improving the quality of "innovation" and "transformation". As a result, it has demonstrated significant effectiveness in guiding industrial agglomeration, promoting innovation and entrepreneurship, providing a fostering ground for the development of new industries, and ensuring a high-quality development of regional economies. Cooperative innovation has the advantage of achieving complementary resources, reducing innovation risks, and increasing the possibility of innovation success, which represents a significant path to accelerating the high-quality development of science and innovation towns [2].

This concept of science and innovation towns was born as a result of China's new urbanization, with science and technology innovation at its core and the integration of resources as a way to gather science and technology enterprises, financial institutions, research and development institutions, and industrial-related enterprises to jointly produce an open and cooperative innovation and entrepreneurship town. Despite the fact that science and technology innovation towns currently have a good development prospect, they also face a number of challenges, such as convergence of industrial characteristics, difficulties in resource integration, insufficient transformation of achievements, and frequent malicious competition. Consequently, the "catch-up innovation" characterized by technology introduction and imitation has become increasingly weak, and technology enterprises in science and innovation towns lack sufficient independent technology innovation capabilities, so they are not competitive internationally. This results in a lack of external competitiveness. Despite this, technological innovation is a "great adventure" that cannot be supported by human or capital resources [3,4], and it is highly risk-dependent [5], which prevents most SMEs from exploring technological innovation. As a solution to the above dilemma, companies in science and innovation towns generally adopt cooperative innovation strategies to enhance their core competitiveness while promoting sustainable and high-quality development.

The concept of "collaborative innovation" is derived from interorganizational cooperation among high-tech companies. To achieve a win-win situation, it is usually based on the common interests of each party to establish a clear goal of cooperation. Following that, resources will be shared, and complementary advantages will be utilized to create a positive outcome. According to Fusfeld and Hakisch [6], collaborative innovation is an agreement between two or more collaborating entities that is based on a common research objective and has complementary factors. In Fritsch's view [7], cooperative innovation involves the division of labor among various cooperative subjects to achieve a certain goal, and participation in the achievement of that goal can be considered cooperative innovation. Cooperative innovation is defined by UNCA [8] as the process of developing new products with the assistance of other organizations. Liu Dan [9] and Wang Zhong and Huang Ruihua [10] examine cooperative innovation as a looser, more flexible partnership between various cooperative subjects based on resource sharing and complementary advantages, with science and technology innovation as the primary objective. In order to remain competitive in the fierce competition of science and technology innovation towns, cooperative innovation has become an important path for enterprises to develop; however, many of them encounter problems such as weak internal R&D capability, low willingness to cooperate, low efficiency of cooperative innovation, free-rider behavior in the cooperation, and insufficient government regulation. Consequently, cooperative innovation has failed, and a great deal of resources have been wasted.

Consequently, it is important to identify the factors that influence the cooperative innovation of enterprises, to reduce the behavior of free-riders in cooperations, to improve the willingness of enterprises to cooperate and innovate, and to promote the smooth development of cooperative innovation among enterprises in science and innovation towns.

Cooperative innovation is essentially an iteration of both cooperation and conflict among various participating subjects for their own interests. Since evolutionary game theory is based on the assumption of finite rationality, which is more in line with reality than traditional game theory based on perfect rationality, a number of scholars have carried out analytical research by constructing a collaborative innovation interest game model based on the above-mentioned studies with the government, universities, and enterprises as the main subjects [11–18]. Among them, Wang Guohong [11] analyzed the evolutionary mechanism and evolutionary process of cooperative innovation from the perspective of two different strategies, namely, "competitive strategy" and "cooperative strategy". Chen Jin [12] investigated the factors influencing the sustainability and stability of cooperative innovation. Ding Xiaozhou [13] constructed an evolutionary game model for both parties from the perspective of technological differences to investigate the evolutionary paths of

innovation model selection, stabilization strategies, and the mechanism of the effects of their parameters in science and technology innovation enterprises. Sun Kai [14] analyzed the influence of trust, complementarity, and risk coefficient on the stability of cooperative innovation of enterprises based on the competition theory using the Anwar game model. Zhang Fang [16] used an evolutionary game theory and method to analyze the influence of different government support methods on civil–military cooperative innovation to construct an evolutionary game model with three parties: civil enterprises, military enterprises, and government. Wu Jun [17] constructed a game model of cooperative innovation between telecommunication enterprises and Internet enterprises in the context of the hybridization of state-owned enterprises. Parameters such as cost, benefit, and apportionment coefficient were found to have significant effects on the evolutionary path of the system, and a default compensation mechanism was introduced to promote cooperative innovation among enterprises.

In summary, previous scholars have provided useful references on cooperative innovation among enterprises in science and innovation towns, but there are still the following problems: First, as a new government-led cooperative innovation organizational model, including a variety of cooperative innovation organizational forms, there are still very few results of detailed research on science and innovation towns, and few studies have examined the dynamic evolutionary process of corporate cooperative innovation in science and innovation towns as the main body. Second, most studies on enterprise collaborative innovation only consider government subsidies and incentives and do not consider other government regulatory behaviors. Moreover, although some studies establish a tripartite game model between enterprises and the government and treat the government as the game subject of interest in the process of cooperative innovation, in reality, the government mainly plays a regulatory role and does not gain benefits. Therefore, this paper adds government regulation behaviors into the game and considers the impact of government regulations such as subsidy incentives and punishment mechanisms on enterprise cooperative innovation. In view of this, this paper uses evolutionary game theory to construct a game model of cooperative innovation among science and innovation town enterprises, discusses the dynamic evolutionary stabilization strategy of cooperative innovation among science and innovation town enterprises under market mechanisms and government regulations by situation, and analyzes the factors affecting the choice of a cooperative innovation strategy for science and innovation town enterprises through MATLAB numerical simulation to verify, so as to promote the sustainable and theoretical basis for policy suggestions, which are provided to promote the sustainable and high-quality development of science and innovation towns in China.

The rest of the paper is organized as follows: Section 2 reviews the literature related to enterprise cooperative innovation, and Section 3 presents the basic hypotheses. Section 4 establishes a model for the evolution of enterprise cooperative innovation under a market mechanism, solves the model, and analyzes the evolutionary trends of enterprise cooperative innovation in science and innovation towns under different conditions. Section 5 establishes a model of cooperative innovation under government regulation, solves the model, and analyzes the evolutionary trends of cooperative innovation among enterprises in a science and technology town under different conditions. Section 6 conducts a numerical simulation to verify the accuracy of the evolutionary game model. The last part is the conclusion and countermeasure suggestions.

## 2. Literature Background

### 2.1. Enterprise Technology Innovation Challenges

The innovation of a firm is a "great adventure" for it, and there are various factors that prevent it from innovating. Gillian Barrett [19] found that the management style of the firm's founder has a significant impact on its innovation. Therefore, the lack of confidence and entrepreneurship of the founder is the most important obstacle to the development of technological innovations. In his study, Alberto Bertello [20] analyzed qualitative data

and identified four stages of technological innovation challenges for SMEs, including planning, implementation, closing, and monitoring. According to Barbara Bigliardi [21], knowledge, organization, collaboration, finance, and strategy are the five main barriers to innovation in SMEs. SMEs from traditionally less innovative industries, along with their clusters, were considered to be important organizational and collaborative barriers to open innovation adoption. As a possible solution to facilitate the project's success, Ullrich and Vladova [22] proposed a framework of trade-offs and decision-making processes. The most effective way to encourage a firm's technological innovation is to compare the positive and negative aspects of the firm's innovation in a comprehensive way. Most studies focus on the positive aspects, while neglecting the negative aspects. According to Nabil Amara [23], more information sources for firms can increase their motivation for innovation, based on the 1999 Statistics Canada Innovation Survey. Technology innovations are better assessed when firms have access to diverse and comprehensive information.

### 2.2. Corporate Collaborative Innovation Factors

#### 2.2.1. The Company's Own Factor

Collaboration can be greatly influenced by a firm's resource endowment and its own characteristics. Whether enterprises adopt cooperative innovation is directly affected by the characteristics of their CEOs, and enterprises with CEOs who are not afraid of failure are more likely to do so [24]. An important criterion for selecting a partner is the ability of the enterprise to conduct R&D, while the enterprise's level of innovation technology and mode of managing partnerships help establish the foundation of cooperative innovation, as well as enhance the level and improve the performance of cooperative innovation [25]. Knowledge is the core resource in cooperative innovation activities, and the more knowledge, the more liquid and valuable it is, the deeper the degree of cooperation can be [26]. A company's search ability is categorized into external search ability and internal search ability, both of which are beneficial to the accomplishment of cooperative innovation activities [27]. However, enterprises with a strong external search ability are more helpful in establishing a good reputation outside of the company and promoting cooperative innovation. As a result of their past experiences with collaborative innovation, companies with a wealth of collaborative innovation experience will be more proactive and make better contributions to the achievement of collaborative goals and reduce the risks associated with accidents [28].

#### 2.2.2. Relationships between Companies

There are also some scholars who analyze the impact of the distribution of costs and benefits of cooperative innovation among enterprises, the selection of partners, the willingness to cooperate, and other factors on the development of cooperative innovation among enterprises. The distribution of enterprise cooperative costs and benefits has the greatest impact on enterprise cooperative innovation, which is designed to achieve or not achieve the goal of enterprise cooperative innovation [29]. Cooperative innovation requires good partners, sticky demand, and heterogeneity of information, which are the decisive factors for enterprise cooperative innovation, as well as partners with heterogeneity and complementarity [30]. Cooperate innovation involves the transfer of knowledge, and explicit knowledge is closely related to an enterprise's risk taking, while invisible knowledge is closely related to the enterprise's trust [31]. A contractual arrangement plays a pivotal role in corporate collaborative innovation, and the contract itself cannot promote successful collaboration, but it can build good trust during the negotiation process [32]. As noted by René Belderbos [33], continuous and stable cooperation is beneficial for improving cooperative innovation performance, as he studied three temporal dimensions of cooperative innovation relationships: initial specific cooperation, long-term cooperation, and interrupted cooperation.

### 2.2.3. The External Environment

A firm's external environment is also considered to be an important factor in influencing collaborative innovation. Swink [34] divides the external and internal environments affecting collaborative innovation into two categories, identifying the external environment as the fickle market competition, among others, and concluding that it is the internal environment that is the main factor affecting innovation performance. Collaboration innovation decisions are influenced greatly by the conceptualization of the organizational environment [35]. Furthermore, it has been suggested that the more competitive the industry is, the greater the possibility of collaboration and the greater the frequency of collaboration [36]. Collaborative innovation may not be directly affected by external uncertainties, but they may moderate the process [37]. Subsidies from government agencies are also included in the external environment factor. Fölster, Stefan [38] found in a study of 45 technology competitions that subsidies are only significant when the results of collaborative innovation must be shared, and they are not significant if the results of collaborative innovation need not be shared. Using the system dynamics theory, Li, Lin [39] concluded that dynamic governmental punishment mechanisms are effective in promoting collaborative innovation among firms.

## 3. Basic Assumptions and Model Building

### 3.1. Corporate Cooperation and Innovation Game Relationship

In its essence, the process of inter-firm cooperative innovation is a game of cooperation and innovation between enterprises in pursuit of maximizing their own interests [40]. Due to the limitations of the innovation subjects' own conditions, cognitive ability, social environment, and resources, enterprises are finitely rational and do not find the optimal strategy at the beginning but will gradually stabilize to the optimal one after many games in the process of continuous learning and trying. The evolutionary game is an analysis method that combines game analysis with a dynamic evolutionary process, and the equilibrium strategy of the game's parties is constantly adjusted and improved instead of being the result of a one-time selection [41]. Therefore, this paper uses evolutionary game theory to analyze the cooperative innovation behavior of firms. Firstly, an evolutionary model of cooperative innovation with and without government regulation is constructed, and the parameters of the model are solved; then, the stable state of the cooperative innovation system under different initial conditions is verified through simulation experiments; finally, some suggestions for promoting the development of cooperative innovation are proposed based on the conclusions.

### 3.2. Model Construction

#### 3.2.1. Gaming Sides

The innovation subjects of this paper, Enterprise A and Enterprise B, are two independent and complementary science and technology enterprises randomly selected from the total group of enterprises in science- and technology-inspired characteristic towns, and they have complementary factors, which, in a narrow sense, means that the value of the enterprise's products, technologies, etc. must be combined with specific products, assets, or technologies of other enterprises to create or realize [42]. Both achieve R&D cooperation on the basis of reasonable profit sharing and cost sharing [43]. When both firms cooperate, they can choose to collaborate and innovate by selecting models such as simple cooperation of existing products, personalization, and co-development of cutting-edge technologies according to market demand. The complementarity of the two is the key for both enterprises to reach cooperative innovation and promote the healthy and sustainable development of science- and innovation-based characteristic towns while promoting their own development. First of all, based on technological innovation, both companies can transform their existing technologies or innovate cooperatively to adapt to market changes through complementary advantages, thus generating value-added technology, from which both companies can gain excess revenue, but the size of the revenue depends on the coefficient

of effort and the willingness of each company to innovate cooperatively. Secondly, both companies choose to cooperate in innovation to achieve cost sharing and risk sharing [44], but the size of the cost of cooperation is also related to the enthusiasm and commitment of science and innovation companies to participate in cooperative innovation. Again, the externality of science and innovation enterprises causes technology spillover between and within enterprises, and there is a stronger technology mobility among the subjects who choose cooperative innovation [45]. Even if a confidentiality agreement between subjects can temporarily suppress technology spillover to the environment outside the subjects to a certain extent, there is still a possibility that one party will follow or imitate the innovation and adopt forward integration or backward integration [46] or cause subjective and objective information leakage due to malicious competition, "free-riding" [47], and other unethical behaviors or other circumstances. The information leakage caused by malicious competition, "free-riding", or other circumstances may create the risk of imitation and copying of technology for the party that chooses to collaborate, resulting in a reduction in the benefits of collaboration. However, with the improvement of the legal and regulatory system, enterprises will pay a heavier price in the case of unethical behavior, which in turn will have a certain inhibiting effect on the prospective gains and losses. Finally, due to the contemporary significance and strategic attributes of science- and innovation-based characteristic towns, the government will actively participate in macroeconomic control when the market mechanism fails and will take into account the social benefits of cooperative innovation when formulating rewards and penalties and indirectly guide both companies involved in the game to work together to achieve technological innovation.

### 3.2.2. Game Strategy

In the process of the cooperative innovation game between enterprises in the science and innovation town, the strategic space of the two main bodies of Enterprise A and Enterprise B is (cooperative innovation, non-cooperative innovation). The strategy of "non-cooperative innovation" means that both parties of the enterprise fail to reach a cooperation intention or that one of the parties defaults and withdraws after reaching a cooperation intention, and the cooperative innovation fails to proceed smoothly.

### 3.2.3. Model Assumptions

**Hypothesis 1 (H1).** *The initial returns of Enterprise A and Enterprise B are $R_1$ and $R_2$, respectively.*

**Hypothesis 2 (H2).** *If both companies choose to cooperate in innovation, there will inevitably be a certain amount of human, financial, and material cost input [48]; the total cost is recorded as C, where the cost sharing rate of company A is $\beta$, and the cost sharing rate of company B is $1 - \beta$, $\beta \in [0, 1]$.*

**Hypothesis 3 (H3).** *If both companies choose cooperative innovation, they can obtain cooperative innovation benefits, and p denotes cooperative innovation benefits.*

**Hypothesis 4 (H4).** *The excess benefit obtained by cooperative innovation is proportional to the coefficient of willingness to cooperate and the coefficient of cooperative innovation effort of both enterprises [49]. The cooperative willingness coefficient of the cooperative innovation of enterprises A and B is $a_i(i = 1, 2)$, and the cooperative innovation effort coefficient is $f_i(i = 1, 2)$, that is, the enterprise obtains an excess return of $a_i f_i p (i = 1, 2)$.*

**Hypothesis 5 (H5).** *Due to the unbalanced resource endowment and different pursuit goals among the enterprises in the characteristic science and innovation towns, the cooperation problems of enterprises participating in cooperative innovation are very complicated, and the phenomenon of mutual "free rider" is also common [50]. This leads to a decline in the innovation income of high-input and high-output entities, while the innovation income of low-input and low-output*

*enterprises increases unreasonably. Here, it is assumed that k is the increase or decrease in the collaborative innovation income k of both companies; when there is a "free-rider" behavior, k > 0.*

**Hypothesis 6 (H6).** *In the process of the collaborative innovation game between enterprises in the science and innovation town, the probability of Enterprise A choosing cooperative innovation is x, and the probability of choosing non-cooperative innovation is* $1 - x$. *The probability of Enterprise B choosing cooperative innovation is y, and the probability of choosing not to cooperate is y. The probability of innovation is* $1 - y$, *and* $x, y \in [0, 1]$.

## 4. An Evolutionary Game Model of Cooperative Innovation under the Market Mechanism

Based on the above basic assumptions, the payoff matrix of the strategy selection game of Enterprise A and Enterprise B can be obtained when the government does not supervise; see Table 1 for details.

**Table 1.** Game matrix of enterprises under market mechanism.

|  |  | Enterprise B | |
| --- | --- | --- | --- |
|  |  | Cooperative innovation $(y)$ | $\mathrm{non-cooperative\ innovation}\ (1-y)$ |
|  |  | $R_1 + a_1 f_1 p - C\beta$ | $R_1 - C\beta - k$ |
|  | Cooperative innovation $(x)$ | $R_2 + a_2 f_2 p - C(1-\beta)$ | $R_2 + k$ |
| Enterprise A |  | $R_1 + k$ | $R_1$ |
|  | non-cooperative innovation $(1-x)$ | $R_2 - C(1-\beta) - k$ | $R_2$ |

In order to find out the dynamic process of replication of the game between Enterprise A and Enterprise B, according to the relevant theory of evolutionary games and the calculation method of expected return, let $E_1$ and $E_2$ represent the average expected return of Enterprise A and Enterprise B, respectively.

The expected returns $E_{11}$, $E_{12}$ and average expected return $E_1$ of Enterprise A choosing cooperative innovation and non-cooperative innovation are:

$$E_{11} = y(R_1 + a_1 f_1 p - C\beta) + (1 - y)(R_1 - C\beta - k) \tag{1}$$

$$E_{12} = y(R_1 + k) + (1 - y)R_1 \tag{2}$$

$$E_1 = xE_{11} + (1-x)E_{12} = x[y(R_1 + a_1 f_1 p - C\beta) + (1-y)(R_1 - C\beta - k)] \\ + (1-x)[y(R_1 + k) + (1-y)R_1] \tag{3}$$

The expected returns $E_{21}$, $E_{22}$ and the average expected return $E_2$ of Enterprise B choosing cooperative innovation and non-cooperative innovation are:

$$E_{21} = x[R_2 + a_2 f_2 p - C(1-\beta)] + (1-x)[R_2 - C(1-\beta) - k] \tag{4}$$

$$E_{22} = x(R_2 + k) + (1 - x)R_2 \tag{5}$$

$$E_1 = yE_{21} + (1-y)E_{22} = y\{x[R_2 + a_2 f_2 p - C(1-\beta)] + (1-x)[R_2 - C(1-\beta) - k]\} \\ + (1-y)[x(R_2 + k) + (1-x)R_2] \tag{6}$$

The replication dynamic equations of firm A and firm B are:

$$F(x) = \frac{dx}{dt} = x(E_{11} - E_1) = x(1-x)(ya_1 f_1 p - C\beta - k) \tag{7}$$

$$F(y) = \frac{dx}{dt} = y(E_{21} - E_2) = y(1-y)[xa_2 f_2 p - C(1-\beta) - k] \tag{8}$$

### 4.1. The Equilibrium Point of the Strategy

The stable state of replication dynamics means that, when the probability of participating in cooperative innovation enterprises choosing two different strategies remains unchanged, and the replication dynamics equation is equal to 0, the entire evolutionary equilibrium point of the system can be obtained. Therefore, let $\frac{dx}{dt} = 0$, $\frac{dy}{dt} = 0$ and obtain the five equilibrium points of the game dynamic system of the cooperative innovation of Enterprise A and Enterprise B: $O(0,0)$, $A(0,1)$, $B(1,1)$, $C(1,0)$, $D(x^*,y^*)$. In addition, $x^* = \frac{C(1-\beta)+k}{a_2 f_2 p}$, $y^* = \frac{C\beta+k}{a_1 f_1 p}$.

### 4.2. Stability Analysis of Equilibrium Point

According to the Friedman method, the evolutionary stability strategy (ESS) of the system is obtained from the stability analysis of the Jacobian matrix of the two-dimensional continuous dynamic system [51]. The Jacobian matrix can be obtained from the calculation of the replicated dynamic equations of Enterprises A and B as:

$$J = \begin{bmatrix} \frac{\partial F(x)}{\partial x} & \frac{\partial F(x)}{\partial y} \\ \frac{\partial F(y)}{\partial x} & \frac{\partial F(y)}{\partial y} \end{bmatrix} = \begin{bmatrix} (1-2x)(ya_1 f_1 p - C\beta - k)x(1-x)a_1 f_1 p \\ y(1-y)a_2 f_2 p(1-2y)[xa_2 f_2 p - C(1-\beta) - k] \end{bmatrix} \tag{9}$$

When the Jacobian matrix satisfies $Det(J) > 0$, $Tr(J) < 0$, the local equilibrium point is the stable strategy of the system. Both $x^*$ and $y^*$ are on the $R = \{(x,y) | 0 \leq x \leq 1, 0 \leq y \leq 1\}$ plane; then, when $0 \leq C(1-\beta) + k \leq a_2 f_2 p$, $0 \leq C\beta + k \leq a_1 f_1 p$ is satisfied, there are five local equilibrium points in the system, as shown in Table 2:

**Table 2.** Stability analysis of local equilibrium points under market mechanism.

| Balance Point | Det(J) | Tr(J) | Stability |
|:---:|:---:|:---:|:---:|
| $O(0,0)$ | + | - | ESS |
| $A(1,0)$ | + | + | Unstable |
| $B(1,1)$ | + | + | ESS |
| $C(1,0)$ | + | - | Unstable |
| $D(x^*,y^*)$ | - | 0 | Saddle Point |

It can be seen from Table 2 that, among the five partial equilibrium points, $O(0,0)$ and $B(1,1)$ are the stable strategy points of Enterprise A and Enterprise B in the characteristic technological innovation town, corresponding to (no innovation cooperation, no innovation cooperation) and (innovation cooperation, innovation cooperation), respectively. Two Pareto optimal results, $A(0,1)$ and $C(1,0)$, are unstable points, and $D(x^*,y^*)$ is a saddle point. From Table 2, the phase diagram of the evolutionary game between the two parties can be drawn, as shown in Figure 1:

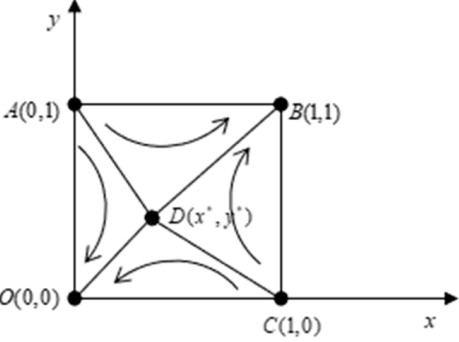

**Figure 1.** Evolutionary phase diagram of both sides of the game.

As shown in Figure 1, the long-term evolution process of Enterprise A and Enterprise B is manifested in that both parties choose "cooperative innovation" or both parties choose a "non-cooperative innovation" strategy. If the initial state falls within the quadrilateral $OADC$, then the system will converge to point $O$ in the long-term evolutionary game, that is, both firm A and firm B choose the strategy of "non-cooperative innovation"; if the initial state falls within the quadrilateral $ABCD$, then the system will, in the long-term evolutionary game, converge to point $B$, that is, both Enterprise A and Enterprise B choose the "cooperative innovation" strategy.

*4.3. Parametric Analysis*

On this basis, according to the above results, a parameter analysis is carried out on whether the two companies adopt a cooperative innovation strategy based on factors such as the cost of cooperative innovation and the willingness of cooperative innovation. The final evolution result of the evolutionary game of both companies depends on the area of quadrilateral $OADC$ and the area of quadrilateral $ABCD$. When $S_{OADC} > S_{ABCD}$, both companies are more likely to choose the "non-cooperative innovation" strategy; when $S_{OADC} < S_{ABCD}$, the two companies are more likely to choose the "cooperative innovation" strategy; when $S_{OADC} = S_{ABCD}$, the evolutionary system will combine the two strategies. The possibility of evolution is the same. Therefore, if we want to analyze the choice of cooperation strategies between the two companies, we can infer the evolution direction of the system by analyzing the various factors that affect the change of $S_{OADC}$ from the perspective of affecting the area of the quadrilateral $OADC$.

**Proposition 1.** *Under the market mechanism, the probability that both companies choose cooperative innovation decreases as the cost of cooperation increases.*

**Proof.** $\frac{\partial S_{OADC}}{\partial C} = \frac{1}{2}\left(\frac{1-\beta}{a_2 f_2 p} + \frac{\beta}{a_1 f_1 p}\right)$, since the income of cooperative innovation between the two parties must be greater than 0, otherwise there is no possibility of cooperation; and because $\beta \in [0,1]$, so $\frac{\partial S_{OADC}}{\partial C} > 0$, so $S_{OADC}$ is a monotonically increasing function of $C$, that is, $S_{OADC}$ increases with the cost of cooperative innovation input by both parties in the game. If it increases, the area of $S_{ABCD}$ will decrease, the probability of the system evolving to $O(0,0)$ will increase, and both sides of the enterprise are more inclined to choose not to cooperate in innovation. Most of the enterprises in the characteristic scientific and technological innovation town are in the early stage of the development of emerging industries, the investment amount is large, and the cost recovery period is long. Therefore, when the cost of cooperative innovation exceeds the expectations of both parties, the enterprises will not carry out cooperative innovation. □

**Proposition 2.** *Under the market mechanism, the probability that both companies choose cooperative innovation increases with the increase in cooperative innovation benefits.*

**Proof.** $\frac{\partial S_{OADC}}{\partial p} = \frac{1}{2}\left(\frac{-a_2 f_2 [C(1-\beta)+k]}{(a_2 f_2 p)^2} - \frac{a_1 f_1 (C\beta+k)}{(a_1 f_1 p)^2}\right) < 0$; therefore, $S_{OADC}$ is a monotonically decreasing function of $p$, that is, $S_{OADC}$ decreases with the increase in the income of cooperative innovation of both sides of the game, and the area of $S_{ABCD}$ will become larger; thus, the probability of the system evolving to $B(1,1)$ will increase, and both sides of the enterprise are more inclined to cooperative innovation. The enterprise itself is mainly profit-making. The greater the benefits obtained by both parties through cooperative innovation, the more the two parties will strive to achieve a willingness to cooperate and jointly obtain greater benefits. □

**Proposition 3.** *Under the market mechanism, the probability of both companies choosing cooperative innovation increases with the increase in the coefficient of willingness to cooperate and the coefficient of cooperative innovation effort.*

**Proof.** $\frac{\partial S_{OADC}}{\partial a_2} = \frac{1}{2}\frac{-pf_2[C(1-\beta)+k]}{(a_2f_2p)^2}$, $\frac{\partial S_{OADC}}{\partial f_2} = \frac{1}{2}\frac{-pa_2[C(1-\beta)+k]}{(a_2f_2p)^2}$; because $-pa_2[C(1-\beta)+k] < 0$, so $\frac{\partial S_{OADC}}{\partial a_2} < 0$, $\frac{\partial S_{OADC}}{\partial f_2} < 0$, similarly, $\frac{\partial S_{OADC}}{\partial a_1} < 0$, $\frac{\partial S_{OADC}}{\partial f_1} < 0$. Therefore, $S_{OADC}$ is a monotonically decreasing function of $a$ and $f$, that is, $S_{OADC}$ decreases with the increase in the cooperative innovation willingness coefficient and effort coefficient of both sides of the game; then, the area of $S_{ABCD}$ will become larger, and the probability of the system evolving to $B(1,1)$ will increase. If the enterprises of both sides trust each other and are willing to work together, the two sides will definitely choose cooperation and innovation. □

**Proposition 4.** *Under the market mechanism, the greater the synergistic benefit of free-riding, the more the two companies tend not to cooperate in innovation strategies.*

**Proof.** $\frac{\partial S_{OADC}}{\partial k} = \frac{1}{2}(\frac{1}{a_2f_2p} + \frac{1}{a_1f_1p})$; because $a_2f_2p > 0$, $a_1f_1p > 0$, so $\frac{\partial S_{OADC}}{\partial k} > 0$, so $S_{OADC}$ is a monotonically increasing function of $k$, that is, $S_{OADC}$ increases with the increase in the cooperative innovation willingness coefficient and effort coefficient of both sides of the game; then, the area of $S_{ABCD}$ will become smaller, and the probability that the system evolves to $O(0,0)$ increases. Enterprises that choose cooperative innovation have to bear the costs and risks of innovation, while companies that do not choose cooperative innovation can share benefits through free-riding. Therefore, both sides of the game hope that the other party will choose a cooperative innovation strategy, so that they can obtain "free-rider" benefits. With the increase or decrease in collaborative innovation benefits due to "free-rider" behavior, the system tends to evolve in the direction of non-cooperative innovation. □

## 5. Considering Enterprise Cooperative Innovation under Government Regulation

The healthy development of characteristic science and technology towns is of great strategic significance for promoting my country's new urbanization process and implementing innovative development strategies [52]. The new features and new potentials of reforming, cultivating, and developing emerging industries and realizing regional economic transformation have become the focus of increasing attention. However, due to factors such as lack of funds, high technical thresholds, talent shortages, and insufficient intellectual property protection, no effective inter-industry collaborative innovation system has been established in all regions. In addition, this is the key to building an "innovative country". Therefore, in order to promote inter-industry cooperation and innovation, local governments actively guide inter-industry cooperation, provide sufficient policy guarantees in government subsidies, equity investment, loan discounts, risk compensation, etc., and continuously create a good regulatory environment for inter-industry cooperation and innovation [53,54].

**Hypothesis 7 (H7).** *The government will bring subsidy incentives and punishment mechanisms to encourage the cooperative innovation of enterprises in scientific and technological innovation-oriented towns. For enterprises that choose cooperative innovation, the government will provide cost subsidies, and the cost subsidies obtained by enterprises are mC. Among them, m is the cost subsidy intensity, $m \in [0,1]$, and C is the total cost of cooperative innovation. In addition, in order to promote enterprise cooperation and innovation, the government will also issue incentive bonuses H to enterprises participating in cooperative innovation. At the same time, in order to achieve effective allocation of innovation resources and fair competition among enterprises, the government imposes a Class F fine on enterprises that have reached a cooperation intention but fail to comply with the cooperation intention [55]. The income matrix of both parties is shown in Table 3:*

**Table 3.** The game matrix of enterprises under government regulation.

| | Enterprise B | |
|---|---|---|
| | Cooperative innovation $(y)$ | $non-cooperative\ innovation\ (1-y)$ |
| | $R_1 + a_1 f_1 p - C\beta + mC + H$ | $R_1 - C\beta - k + mC + H$ |
| Cooperative innovation $(x)$ | $R_2 + a_2 f_2 p - C(1-\beta) + mC + H$ | $R_2 + K - F$ |
| Enterprise A | $R_1 + K - F$ | $R_1$ |
| non-cooperative innovation $(1-x)$ | $R_2 - C(1-\beta) - k + mC + H$ | $R_2$ |

The expected benefits $E'_{11}$, $E'_{12}$ and the average expected benefit $E'_1$ of Enterprise A choosing cooperative innovation and non-cooperative innovation are:

$$E'_{11} = y(R_1 + a_1 f_1 p - C\beta + mC + H) + (1-y)(R_1 - C\beta - k + mC + H) \quad (10)$$

$$E'_{12} = y(R_1 + K - F) + (1-y)R_1 \quad (11)$$

$$E'_1 = xE'_{11} + (1-x)E'_{12} = x[y(R_1 + a_1 f_1 p - C\beta + mC + H) + (1-y)(R_1 - C\beta - k + mC + H)] \\ +(1-x)[y(R_1 + K - F) + (1-y)R_1] \quad (12)$$

The expected benefits $E'_{21}$, $E'_{22}$ and the average expected benefit $E'_2$ of Enterprise B choosing cooperative innovation and non-cooperative innovation are:

$$E'_{21} = x[R_2 + a_2 f_2 p - C(1-\beta) + mC + H] + (1-x)[R_2 - C(1-\beta) - k + mC + H] \quad (13)$$

$$E'_{22} = x(R_2 + k - F) + (1-x)R_2 \quad (14)$$

$$E'_2 = yE'_{21} + (1-y)E'_{22} = y\{x[R_2 + a_2 f_2 p - C(1-\beta) + mC + H] \\ +(1-x)[R_2 - C(1-\beta) - k + mC + H)]\} + (1-y)[x(R_2 + k - F) + (1-x)R_2] \quad (15)$$

The replication dynamic equations of firm A and firm B are:

$$F'(x) = \frac{dx}{dt} = x(E'_{11} - E'_1) = x(1-x)[y(a_1 f_1 p + F) - C\beta - k + mC + H] \quad (16)$$

$$F'(y) = \frac{dx}{dt} = x(E'_{21} - E'_2) = y(1-y)[x(a_2 f_2 p + F) - C(1-\beta) - k + mC + H] \quad (17)$$

### 5.1. Equilibrium Point of the Strategy

The stable state of replication dynamics refers to the level at which the proportion of players who adopt two strategies in the group remains unchanged. If the replication dynamics equation is equal to 0, the entire evolutionary equilibrium point of the system can be obtained. Therefore, set $\frac{dx}{dt} = 0$, $\frac{dy}{dt} = 0$, and obtain five equilibrium points of the game dynamic system of cooperative innovation between Enterprise A and Enterprise B: $O'(0,0)$, $A'(0,1)$, $B'(1,1)$, $C'(1,0)$, $D'(x^*, y^*)$. In addition, $x^* = \frac{C(1-\beta)+k-mC-H}{a_2 f_2 p + F}$, $y^* = \frac{C\beta+k-mC-H}{a_1 f_1 p + F}$.

### 5.2. Stability Analysis of Equilibrium Point

According to the Friedman method, the evolutionary stability strategy (ESS) of the system is obtained from the stability analysis of the Jacobian matrix of the two-dimensional continuous dynamic system, and the Jacobian matrix can be obtained from the replication dynamic equations of Enterprise A and Enterprise B as:

$$J = \begin{bmatrix} \frac{\partial F'(x)}{\partial x} & \frac{\partial F'(x)}{\partial y} \\ \frac{\partial F'(y)}{\partial x} & \frac{\partial F'(y)}{\partial y} \end{bmatrix} = \begin{bmatrix} (1-2x)[y(a_1 f_1 p + F) - C\beta - k + mC + H] & x(1-x)(a_1 f_1 p + F) \\ y(1-y)(a_2 f_2 p + F) & (1-2y)[x(a_2 f_2 p + F) - C(1-\beta) - k + mC + H] \end{bmatrix} \quad (18)$$

When the Jacobian matrix satisfies $Det(J) > 0$, $Tr(J) < 0$, the local equilibrium point is the stable strategy of the system. Both $x^*$ and $y^*$ are on the $R = \{(x,y)|0 \le x \le 1, 0 \le y \le 1\}$

plane; then, when $0 \leq C(1-\beta) + k - H \leq a_2 f_2 p + F$ and $0 \leq C\beta + k - H \leq a_1 f_1 p + F$ are satisfied, there are five local equilibrium points in the system, as shown in Table 4:

**Table 4.** Stability analysis of local equilibrium points under government regulation.

| Balance Point | $Det(J)$ | $Tr(J)$ | Stability |
|---|---|---|---|
| $O'(0,0)$ | + | - | ESS |
| $A'(0,1)$ | + | + | Unstable |
| $B'(1,1)$ | + | + | ESS |
| $C'(1,0)$ | + | - | Unstable |
| $D'(x^*, y^*)$ | - | 0 | Saddle Point |

It can be seen from Table 4 that, among the five partial equilibrium points, $O'(0,0)$ and $B'(1,1)$ are the stable strategy points of Enterprise A and Enterprise B in the characteristic science and technology innovation town under government regulation, corresponding to (no innovation cooperation, no innovation cooperation) and (innovation cooperation, innovation cooperation). Two kinds of Pareto optimal results, $A'(0,1)$ and $C'(1,0)$, are unstable points, and $D'(x^*, y^*)$ is a saddle point.

*5.3. Parametric Analysis*

From Table 4, the phase diagram of the evolutionary game between the two parties that can be drawn is the same as that in Figure 1. Whether the unstable point will eventually move to point $O'(0,0)$ or whether point $B'(1,1)$ tends to be stable still depends on the area $S_{O'A'D'C'}$ of the quadrilateral $O'A'D'C'$.

$$S_{O'A'D'C'} = \frac{1}{2}(x^* + y^*) = \frac{1}{2}\left( \frac{C(1-\beta) + k - mC - H}{a_2 f_2 p + F} + \frac{C\beta + k - mC - H}{a_1 f_1 p + F} \right) \quad (19)$$

**Proposition 5.** *Under the government's regulation, with the increase in government subsidies and incentive bonuses, the enterprise obtains more additional income, and the greater the probability that both sides of the enterprise choose to cooperate in innovation.*

**Proof.** $\frac{\partial S_{O'A'D'C'}}{\partial H} = \frac{1}{2}\left( \frac{-1}{a_2 f_2 p + F} - \frac{1}{a_1 f_1 p + F} \right)$; because $a_2 f_2 p + F > 0$, $a_1 f_1 p + F > 0$, so $\frac{\partial S_{O'A'D'C'}}{\partial H} < 0$, and for the same reason, $\frac{\partial S_{O'A'D'C'}}{\partial m} < 0$. Therefore, $S_{O'A'D'C'}$ is a monotonically decreasing function of $H$, that is, $S_{O'A'D'C'}$ decreases with the increase in government subsidies and incentive bonuses; then, the area of $S_{A'B'C'D'}$ will become larger, and the probability of the system evolving to $B'(1,1)$ will increase, and both companies are more inclined to cooperate and innovate. The healthy development of enterprises in the characteristic scientific and technological innovation town is of great historical significance. Therefore, whether through cost subsidies or achievement rewards, the government can effectively reduce the R&D costs of enterprises and encourage enterprises to choose cooperative innovation. □

**Proposition 6.** *Under the government's regulation, the more fines and penalties are imposed on companies that have reached a cooperation intention but breach the contract, the greater the probability that both companies choose to cooperate and innovate.*

**Proof.** Compared with times when the government does not regulate, the difference between $D'$ and $D$ is $H$ in the numerator and $F$ in the denominator. Therefore, according to $H$ and $F$, we can analyze the impact of government regulation on the cooperation and innovation of enterprises in the characteristic scientific and technological innovation town:

(1) When $C(1-\beta) + k > H + mC$ and $c\beta + k > H + mC$, that is, $\frac{C\beta + k - mC - H}{a_1 f_1 p + F} < \frac{c\beta + k}{a_1 f_1 p}$, $\frac{C(1-\beta) + k - mC - H}{a_2 f_2 p + F} < \frac{c(1-\beta) + k}{a_2 f_2 p}$, the area of $S_{O'A'D'C'}$ will become smaller at this time, indicating that when government regulation is introduced again, the probability of the

system evolving to $B'(1, 1)$ increases, and the greater the $F$, the greater the possibility. It shows that the larger the fine imposed by the government on the enterprise, the more inclined the enterprise is to choose cooperative innovation.

(2) When $C(1 - \beta) + k < H + mC$ and $C\beta + k < H + mC$, that is, the additional benefits obtained by the government's subsidies to enterprises that do not participate in cooperative innovation are greater than the cost of cooperation, due to the government's intervention, both companies will choose a cooperative innovation strategy. Co-innovation will gain economic benefits, which in turn will encourage enterprises to actively participate in co-innovation. □

## 6. Numerical Simulation

After demonstrating the influence of the changes of various elements on the strategic choice of the game subject, the evolutionary law of whether to choose cooperative innovation based on the cooperative innovation of the science and technology town was further analyzed. In this paper, Matlab software was used for simulation, and the parameters of the model built in this paper were assigned values based on research on the influencing factors of enterprise cooperative innovation in the related literature [11–18] and on the opinions of experts in related fields.

(1) The total cost of cooperative innovation between Enterprise A and Enterprise B is $C$ = CNY 200,000, the cost sharing rate of enterprise A is $\beta$ = 0.5, and the cost sharing rate of enterprise B is $1 - \beta$ = 0.5.

(2) If both parties choose co-innovation, the total revenue of co-innovation is $p$ = 1.05 million.

(3) The cooperative innovation coefficients of Enterprises A and B are $a_1 = 0.5$ and $a_2 = 0.5$, respectively, and the cooperative innovation effort coefficients of Enterprises A and B are $f_1 = 0.5$ and $f_2 = 0.5$, respectively.

(4) When there is a "free-rider" behavior, the increase or decrease in the collaborative innovation benefit of both companies is $k$ = CNY 50,000.

(5) In order to encourage the cooperation and innovation of enterprises in the characteristic scientific and technological innovation town, the government provides a subsidy of $m$ = 0.05, an incentive bonus $H$ = CNY 10,000, and a fine of $F$ = CNY 10,000 for cooperation and breach of contract.

### 6.1. Simulation Research on Evolutionary Game between Enterprises under the Market Mechanism

6.1.1. Influence of Cooperative Innovation Costs and Benefits on the Evolutionary Results of Both Game Parties

The establishment of a stable cooperative innovation relationship between enterprises is established on the premise of ensuring sufficient benefits. Therefore, the cooperation cost and income of enterprises will ultimately affect the choice of strategies of both parties. In order to verify the total cost and the effect of total revenue on the system's evolutionary result:

(1) From the parameter analysis above, it can be known that the smaller the total cost of enterprise cooperative innovation, the more likely it is to promote the probability that both parties choose cooperative innovation. Therefore, this paper considers that, under the market mechanism, assuming other parameters remain unchanged, with the total cost of enterprise cooperative innovation $C$ ($C = 20, 18, 16, 14$), the impact of the total cost of enterprise cooperative innovation on the system's evolutionary results is shown in Figure 2.

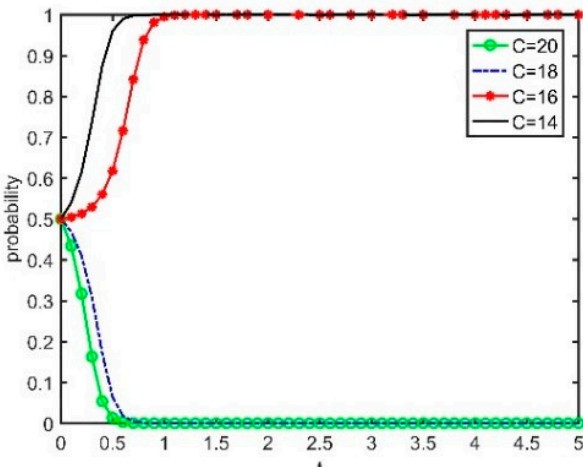

**Figure 2.** Impact of different total costs of collaborative innovation on system's evolutionary outcomes.

From the results in Figure 2, it can be seen that the evolutionary results of the cooperative innovation strategy choices of enterprises in the characteristic science and technology towns are affected by the total cost of cooperative innovation.

(2) From the above parameter analysis, it can be known that the greater the total revenue of enterprise cooperative innovation, the more likely it is to promote the probability that both parties choose cooperative innovation. Therefore, this paper considers that, under the market mechanism, assuming that other parameters remain unchanged, as the enterprise cooperative innovation benefits increase $p$ ($p = 105, 115, 125, 135$), the influence of the total cost of enterprise cooperative innovation on the system's evolutionary results is shown in Figure 3.

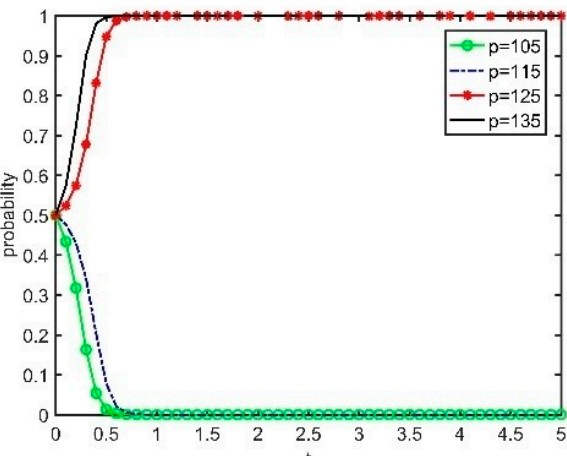

**Figure 3.** Effect of different innovation cooperation benefits on system's evolutionary outcomes.

From the results in Figure 3, it can be seen that the evolutionary results of the cooperative innovation strategy choice of enterprises in the characteristic science and technology towns are affected by the cooperative innovation benefits.

Through the above analysis, the reduction in innovation cooperation cost and the in-crease in income of enterprises can exert their respective advantages to promote the choice of innovation cooperation strategy by both sides of the enterprise, and the income is more sensitive to the impact of the enterprise than the cost. Therefore, both sides of the enter-prise should establish an open, cooperative, and shared knowledge platform so as to ob-tain more cooperative innovation benefits and reinforce corporate cooperation through co-operative innovation benefits.

### 6.1.2. The Influence of Cooperative Willingness Coefficient and Cooperative Effort Coefficient on the Evolution Results of Both Game Parties

The enterprise cooperation willingness coefficient and the enterprise cooperation effort coefficient directly affect the respective final cooperation benefits of the enterprises. Therefore, when $p = 105$, $C = 20$ (the result of the system's evolution is non-cooperative innovation), the effect of the enterprise cooperation willingness coefficient and the cooperation effort coefficient on the system's evolution was further explored.

It can be seen from the above parameter analysis that the stronger the willingness of enterprises to cooperate, the greater the cooperation benefits obtained by the enterprise itself, and the more inclined the enterprise is to choose innovative cooperation strategies. With the increase in Enterprise A's willingness to cooperate $a_1$ ($a_1 = 0.5, 0.6, 0.7, 0.8$), the effect of enterprise cooperation willingness on the system's evolutionary results is shown in Figure 4:

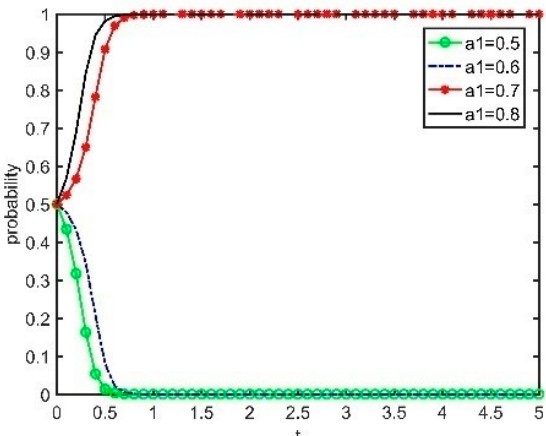

**Figure 4.** The effect of firms' willingness to cooperate on the evolutionary outcome of the system.

From the results in Figure 4, it can be seen that the evolutionary results of the cooperative innovation strategy choice of enterprises in the characteristic science and technology towns are affected by the willingness to cooperate. There is a threshold between 0.6 and 0.7. When the coefficient of willingness to cooperate is greater than the threshold, the enterprise will choose the cooperative innovation strategy. Therefore, the stronger the willingness of enterprises to cooperate and innovate, the more the two sides of the enterprise evolve towards the path of (innovation cooperation, innovation cooperation).

In the same way, the results of the system's evolution of the coefficient of enterprise cooperation effort can be obtained, as shown in Figure 5. The willingness to cooperate is the premise of enterprise cooperation innovation and an important condition for the transformation of achievements. The degree of cooperative effort affects the transformation of results and whether the results of cooperative innovation are achieved. Therefore, it is necessary to fully recognize the importance of the formation and transformation stages of enterprise cooperation and innovation, establish a good knowledge sharing and exchange platform, and enhance the willingness of enterprises to cooperate. In addition, it is necessary to strive to form a cooperative innovation environment that encourages innovation and tolerates failure, maximizes the degree of cooperation between enterprises, and inspires innovation subjects to innovate bravely.

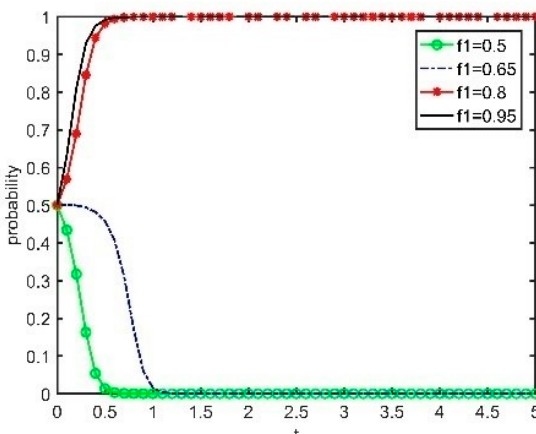

**Figure 5.** Effect of the coefficient of cooperative effort of firms on the evolutionary outcome of the system.

6.1.3. The Influence of "Free-Rider" Behavior on the Evolution Results of Both Sides of the Game

The "free-rider" behavior has a direct impact on the choice of a company's cooperative innovation strategy. In order to explore the impact of the increase or decrease in the collaborative innovation benefits of both companies on the company's cooperative innovation when there is "free-rider" behavior, it is assumed that other parameters remain unchanged. As the synergistic benefit $k$ generated by the "free-rider" behavior decreases ($k$ = 6, 5, 4, 3), the impact of firms' willingness to cooperate on the system evolution results is shown in Figure 6:

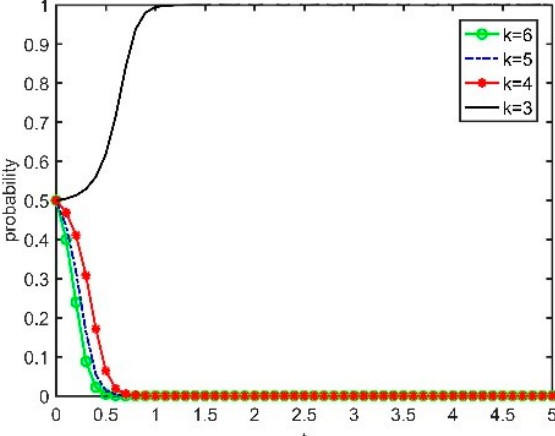

**Figure 6.** Impact of "free-rider" behavior on system's evolutionary outcomes.

It can be seen from Figure 6 that the evolutionary result of the choice of enterprise cooperative innovation strategy in the characteristic science and technology town is affected by "free-rider" behavior. When there is "free-rider" behavior, the increase or decrease in the collaborative innovation benefits of both companies is smaller, and the path evolution of the two sides of the enterprise towards (innovation cooperation, innovation cooperation) is consistent with the result of proposition 4 in the parametric analysis. Due to "free-rider" behavior between enterprises in cooperation, enterprises do not hesitate to sacrifice the interests of other enterprises in order to maximize their own interests, which directly affects the enthusiasm of other enterprises to participate in cooperation and also reduces the effect of cooperation and innovation between enterprises. Therefore, it is necessary to introduce government regulation, through incentive measures such as cost subsidies and bonus incentives for companies that actively participate in cooperative innovation or through

certain penalties for companies that reach a cooperation intention but default, so as to weaken the "free rider" behavior.

### 6.2. Simulation Research on Evolutionary Game of Both Sides of Enterprises under Government Regulation

When the cooperative innovation income $p$ = CNY 1.05 million, the cooperation cost $C$ = CNY 200,000, the cooperation willingness coefficient and the effort coefficient are both 0.5, and the "free-rider" synergistic benefit is CNY 50,000 (the result of the system's evolution is non-cooperative innovation), the government regulation mechanism can be entered to explore its impact on the system's evolutionary results.

#### 6.2.1. The Influence of Government Cost Subsidy Intensity on the Evolution Results of Both Sides of the Game

Only the government cost subsidy was considered, and the incentive bonus was not considered. Assuming that other parameters remain unchanged, the cost subsidy intensity $m$ is introduced. With the increase in the enterprise cost subsidy intensity ($m$ = 0.05, 0.06, 0.07, 0.08), the enterprise cooperation willingness has an influence on the system's evolutionary result. The effect is shown in Figure 7:

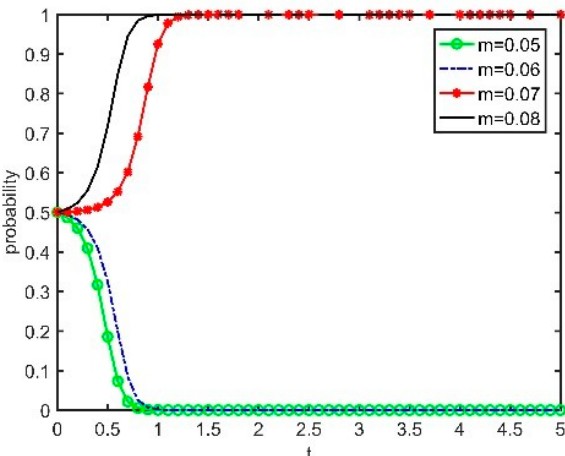

**Figure 7.** Effect of cost subsidy intensity on system's evolutionary outcomes.

As can be seen from Figure 7, the evolutionary results of the choice of enterprise cooperation and innovation strategy in the characteristic science and technology town are affected by the government's cost subsidy, which is consistent with Proposition 5 in the parametric analysis. In addition, there is a threshold for the government's cost subsidy intensity. If the cost subsidy intensity is too small, it will not be able to stimulate the strategic evolution of cooperative innovation in both directions of the game. Therefore, for the sustainable and healthy development of science and technology towns and enterprises, the government should give certain cost subsidies to encourage enterprises to form cooperative innovation alliances and give them positive guidance.

#### 6.2.2. The Influence of Incentive Bonus on the Evolution Results of Both Game Parties

Only the incentive bonus was considered, and the cost subsidy intensity was not considered. Assuming that other parameters remain unchanged, the incentive bonus $H$ is introduced. With the increase in the enterprise incentive bonus ($H$ = 1, 1.5, 2, 2.5), the influence of the enterprise cooperation willingness on the system's evolutionary results is as follows in Figure 8:

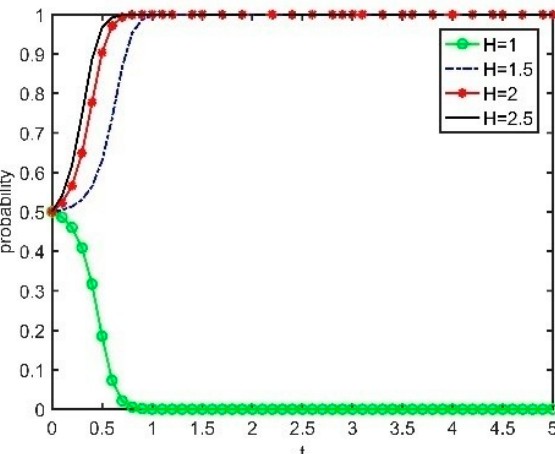

**Figure 8.** The effect of incentive bonuses on the evolutionary outcome of the system.

It can be seen from Figure 8 that the evolutionary results of the choice of enterprise cooperation innovation strategy in the characteristic science and technology town are affected by the government incentive bonus, which also verifies Proposition 5 of the parametric analysis. In addition, it can be seen that, in the early stage of enterprise cooperation and innovation, the incentive effect of the incentive bonus is more obvious. The government should combine the two incentive methods to jointly motivate and promote the enthusiasm of enterprises to cooperate and innovate.

6.2.3. The Influence of Liquidated Penalty on the Evolution Results of Both Game Parties

In order to create a fair, positive, and win-win cooperation and innovation environment, certain penalties are imposed on companies that have reached a willingness to cooperate and then breach the contract. When the cost subsidy is 0 and the incentive bonus is 0, assuming other parameters remain unchanged, a penalty $F$ is introduced, which varies with the increase in fines for defaulting enterprises ($F$ = 1, 1.5, 2, 2.5). The influence of enterprises' willingness to cooperate on the system's evolutionary results is shown in Figure 9:

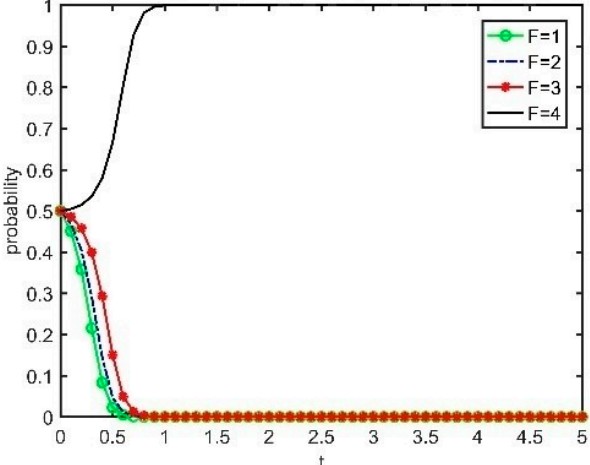

**Figure 9.** The effect of default penalties on the evolutionary outcome of the system.

It can be seen from Figure 9 that the evolutionary results of the choice of enterprise cooperation and innovation strategies in the characteristic science and technology towns are affected by the penalty for breach of contract. This also verifies Proposition 6 of the parametric analysis. It can be seen from the figure that there is a threshold between the fines, and the medium fines have no effect on promoting the cooperative innovation of

enterprises. Only the fines higher than the threshold can avoid the "free-rider" behavior of enterprises in breach of contract and profit. Therefore, in order to ensure the orderly development of enterprise cooperation and innovation, in addition to establishing certain incentives, a penalty higher than the threshold can be set to reduce the "free-rider" behavior and promote enterprise cooperation and innovation.

## 7. Results and Discussion

### 7.1. Conclusions

This paper used the idea and method of an evolutionary game to construct a dynamic evolutionary model of cooperative innovation among enterprises in science- and innovation-oriented characteristic towns under the market mechanism and government regulation, and then used MATLAB to simulate the dynamic evolution of the game of cooperative innovation among innovative enterprises. The dynamic evolution of the cooperative innovation system among the innovative enterprises in the science and innovation town was further analyzed in terms of the benefits and costs of cooperative innovation, the willingness and effort coefficients of cooperative innovation, the speculative behavior of "free-riding", the incentive mechanism of government subsidies, and the penalty for breach of contract. Combined with the previous analysis, the following conclusions can be drawn.

(1) Whether under the market mechanism or government regulation, the cost and benefit of cooperative innovation directly affect the willingness of cooperative innovation of enterprises in a science- and innovation-oriented characteristic town. Moreover, the coefficient of cooperative willingness and the coefficient of cooperative effort of enterprises have positive effects on the evolution of the system tending to the direction of cooperative innovation because they directly affect the benefits obtained by the enterprises' cooperation.

(2) The government can adopt both an incentive bonus and cost subsidy to promote the cooperative innovation of science- and innovation-oriented characteristic town enterprises, which is conducive to improving the stabilization of the system's evolution [56]. At the early stage of cooperative innovation, an incentive bonus is a stronger driver for cooperative innovation, but the incentive effect of a cost subsidy is more durable and efficient at the later stage; the combination of the two incentive methods is more effective than a single incentive method.

(3) The "free-rider" speculative behavior of enterprises can greatly damage the cooperative innovation of enterprises, and the system tends to evolve in the direction of uncooperative innovation as the increase or decrease in synergistic innovation benefits of both enterprises due to "free-rider" behavior becomes larger. At this point, the introduction of the government's penalty regulation mechanism, which imposes fines on enterprises that have reached cooperation intentions and defaulted, can lessen the damage of "free-riding" to the enthusiasm of cooperative innovation among enterprises and promote the system to evolve in the direction of cooperative innovation.

### 7.2. Theoretical Implications

There are three theoretical contributions of our study. First and foremost, our study contributes to the study of characteristic towns by broadening the research population and context. By presenting an evolutionary game model, this study explored the evolution of innovation and development strategies of science- and innovation-oriented characteristic town enterprises from a dynamic perspective, which adds to the literature related to the sustainable and healthy development of science- and innovation-oriented characteristic towns based on previous static perspective studies. Second, through MATLAB simulation analysis, the influence trend of each parameter was systematically presented to explore in more depth the intrinsic influencing factors of the choice of cooperative innovation strategy of enterprises in scientifically and technologically oriented characteristic towns. This will further improve the research on the choice of cooperative innovation strategy for enterprises. Lastly, the literature on the influence of government on innovation focuses mainly on the

influence of national policies on innovation; there is little literature concerning the influence of government punishment mechanisms on enterprise cooperation innovation strategies. It is possible to explore the mechanism by which the government selects different subsidies and penalties on the cooperative innovation of enterprises in science and innovation towns in the absence of the market mechanism, thereby providing a useful exploration of the regulation used by the government to promote the cooperative innovation of enterprises.

*7.3. Suggestions for Countermeasures*

(1) **SMEs are an integral part of the science and innovation town and are an important force in promoting innovation in the town.** Promoting and encouraging the participation of SMEs in collaborative innovation will lay the foundation for the development of science and innovation towns. Enterprises should take advantage of their own advantages and environmental opportunities; enhance their own technological innovation and technological maturity; improve their willingness and efforts to cooperate and innovate; and work on enhancing their creditworthiness, reviewing their long-term growth strategies in due course, cultivating a sense of worry, developing a win-win cooperation vision from a strategic perspective, and working with other SMEs in order to collaborate, share results, and share risks. It is also imperative that SMEs are able to absorb the lessons learned from the success and failure of core enterprises, to learn from the advanced management and technology of core enterprises, to increase their investment in innovation factors, to strengthen the pool of highly skilled and educated workers, and to promote the common progress of all members of the community.

(2) **Build an innovative knowledge platform and consolidate the development platform of science and innovation towns.** It is important that businesses actively participate in the construction of a science- and innovation-based town in order to build it into an open, cooperative, and shared knowledge platform with a clustering effect [57] and to promote technology clustering. As a result of the development of a knowledge sharing platform, clustering and sharing of information, knowledge, and technology will be accelerated. In order to promote the development of a high-quality science and innovation town, the core companies should actively organize and promote its development. Meanwhile, core enterprises must actively organize and drive SMEs and other nodes of the industry chain to cooperate and innovate in order to encourage product development, technology renewal, channel building, and brand building in science and innovation cities.

(3) **Establish a credit collection system and enterprise credibility mechanism to optimize the development ecology of science and innovation towns**. By developing a credit collection system and a credibility mechanism, science and innovation towns will be able to assess the credit status of enterprises that have failed to cooperate and innovate by imitating innovation, malicious competition, and free-riding, among other things. Science and innovation towns and related institutions will boycott companies that do not comply with this requirement in order to encourage them to focus on long-term gains rather than short-term speculative interests in repeated games. Higher credit ratings and policy preferences for reputable firms will enhance win-win cooperation in the repeated game [58], which will further optimize the innovation development ecology of science and innovation towns.

(4) **Enhance the government's role as a scientific guidance organization, and clarify the direction in which science and innovation towns are being developed.** It is imperative that the role of government in the development of a collaborative innovation system is highly emphasized in the process of planning and developing science- and innovation-oriented characteristic towns, as well as the provision of policy support and guidance. Set up government-funded projects and increase investments in science and technology innovation in order to promote the establishment of a systematic science and technology innovation system. Improve the construction of intermediary

services in science- and innovation-oriented characteristic towns to reduce the risk of cooperative innovation among enterprises, while strengthening the protection of intellectual property rights [59] and establishing reasonable financial and incentive mechanisms, as well as moderate punishment mechanisms. By taking advantage of the different sensitivity to incentives and penalties of enterprises participating in cooperative innovation, we should use both financial investment and incentive mechanisms to reduce the cost of innovation subjects participating in cooperative innovation and to effectively enhance the enthusiasm of these enterprises. In addition, it is necessary to establish a moderate punishment mechanism and to formulate a variety of punishment measures in order to motivate more businesses to participate in cooperative innovation in science and innovation towns.

### 7.4. Shortcomings and Prospects

Despite the fact that this study has made some useful explorations, it still has some deficiencies. As a first point, the construction and development of characteristic towns are strongly influenced by Chinese culture, and they require certain policy support and direction for sustainable and healthy development. This paper presents all relevant studies conducted in the context of the Chinese characteristic system, so there are certain limitations related to the Chinese context. Second, this study belongs to the complete information game, and future research can explore ways of encouraging cooperative innovation among enterprises in science- and innovation-oriented characteristic towns with incomplete information. Lastly, since there are many factors affecting the cooperative innovation behaviors of member firms in science- and innovation-oriented characteristic towns, only some important influencing factors have been investigated in this paper; other influencing factors, such as heterogeneity and fairness preferences, have not been considered. It will be necessary to take these factors into account in the future to gain a deeper understanding of the evolutionary process of cooperative innovation behavior among member firms in science and innovation towns.

**Author Contributions:** Conceptualization, F.L. and Y.W.; methodology, F.L.; software, Y.W.; validation, F.L., Y.W.; formal analysis, F.L.; investigation, F.L.; resources, Y.W.; data curation, Y.W.; writing—original draft preparation, Y.W.; writing—review and editing, F.L.; visualization, Y.W.; supervision, F.L.; project administration, F.L.; funding acquisition, F.L. All authors have read and agreed to the published version of the manuscript.

**Funding:** This research was funded by [Beijing Social Science Fund] grant number [19GLB018] And the APC was funded by [Beijing Social Science Fund].

**Institutional Review Board Statement:** Not applicable.

**Informed Consent Statement:** Informed consent was obtained from all subjects involved in the study. Written informed consent has been obtained from the patient(s) to publish this paper.

**Data Availability Statement:** The data that support the findings of this study are available on request from the corresponding author. The data are not publicly available due to privacy or ethical restrictions.

**Conflicts of Interest:** The authors declare no conflict of interest.

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
