# Peer review of "Study on the Evolutionary Game of Cooperation and Innovation in Science and Technology Town Enterprises"

_sustainability, doi:10.3390/su14159210_

Round 1
Reviewer 1 Report
Thank you for the opportunity to read the paper. It is an interesting toping and I consider it fits to the journal. The article reports on a very interesting study that may reach large audiences. The article is very well organized, in conceptual and methodological terms, and presents very relevant results. The research question is clearly stated. The theoretical framework is creative. The research question is explored in a way that is new, creative and important to the discipline. The methodology is clearly explained. The empirical data are analysed in appropriate ways, and written up in ways that are easy to understand. The study conclusions supported are by the analysis. The biography is rich and up-to-date. The authors have done an excellent jo
Reviewer 2 Report
Review report Sustainability 1781121
It is my pleasure to review this manuscript entitled study on the evolutionary game of cooperation and innovation in science and innovation-oriented special town enterprises. Here are major comments
1. some terms are not usual and are not clearly defined. For example, what is innovation-oriented special town enterprises, cooperative innovation, enterprise cooperative innovation. There are many other terms like these, I cannot understand the meaning of them.
2. there is no clear logic in this manuscript. I understand this type of town is driven by policies, why it can be studied by the evolutionary game? For example, what does “Corporate cooperation and innovation game relationship” mean? I have no idea about the innovation games in this context? Who are the players? Enterprises? Why this is relevant to town enterprises? What is the difference between town enterprises and enterprises? It assumes “Enterprise A and Enterprise B are two independent and complementary technology companies in the science and technology town” why do firms have to be complementary?
3. I cannot see any contribution of this manuscript to the literature, it did not explain the key concepts, and cannot define a clear research question. It even did not review literature about innovation cooperation.
4. this manuscript has writing style problems, for example, reference in full name is not a convention. And many other problems. Before the next submission, strongly suggest inviting a native scholar to proofread it.
5. the model does not fit into the reality of innovation cooperation. At least, the authors should review innovation cooperation models and identify research gaps, with those basics, evolutionary models will be possible,
Good Luck with your revision.
Reviewer 3 Report
The authors have conceived a compelling and interesting study. However it needs some improvement:
-It should be stated if the study refers to a specific country and why
- Each hypothesis before formulated should have a background of strong literature review - the article should include at least 50 references
Reviewer 4 Report
Thank you for the opportunity to read your interesting manuscript.
The paper deals with a relevant topic, is well written, well contextualized and the research is well conducted. My main concern is about the “so what” issue.
The economic story is almost all missing and it is difficult to identify the importance and implications of your research.
I suggest to enhance this section and to discuss the research gap you identified, the implications both theoretical and practical including policy implications.
Otherwise it is a well written paper.
Round 2
Reviewer 2 Report
i am not convinced by this revision. you basically should read the works published in Chinese and English, then you may know the frontiers of this research field. Such as research policy, technovation. Just read the articles on firm cooperation in these two journals, you may have a better understanding about your research question.
